# Self-supervised Heterogeneous Graph Pre-training Based on Structural Clustering

**Yaming Yang, Ziyu Guan, Zhe Wang, Wei Zhao,**\* **Cai Xu, Weigang Lu, Jianbin Huang**
School of Computer Science and Technology, Xidian University
{yym@, zyguan@, zwang@stu., ywzhao@mail., cxu@, wglu@stu., jbhuang@}xidian.edu.cn

## Abstract

Recent self-supervised pre-training methods on Heterogeneous Information Networks (HINs) have shown promising competitiveness over traditional semi-supervised Heterogeneous Graph Neural Networks (HGNNs). Unfortunately, their performance heavily depends on careful customization of various strategies for generating high-quality positive examples and negative examples, which notably limits their flexibility and generalization ability. In this work, we present SHGP, a novel Self-supervised Heterogeneous Graph Pre-training approach, which does not need to generate any positive examples or negative examples. It consists of two modules that share the same attention-aggregation scheme. In each iteration, the Att-LPA module produces pseudo-labels through structural clustering, which serve as the self-supervision signals to guide the Att-HGNN module to learn object embeddings and attention coefficients. The two modules can effectively utilize and enhance each other, promoting the model to learn discriminative embeddings. Extensive experiments on four real-world datasets demonstrate the superior effectiveness of SHGP against state-of-the-art unsupervised baselines and even semi-supervised baselines. We release our source code at: `https://github.com/kepsail/SHGP`.

## 1 Introduction

Over the past few years, various semi-supervised graph neural networks (GNNs) have been proposed to learn graph embeddings. They have achieved remarkable success in many graph analytic tasks. This success, however, comes at the cost of a heavy reliance on high-quality supervision labels. In real-world scenarios, labels are usually expensive to acquire, and sometimes even impossible due to privacy concerns.

To relieve the label scarcity issue in (semi-) supervised learning, and take full advantage of a large amount of easily available unlabeled data, the self-supervised learning (SSL) paradigm has recently drawn considerable research interest in the computer vision research community. It leverages the supervision signal from the data itself to learn generalizable embeddings, which are then transferred to various downstream tasks with only a few task-specific labels. One of the most common SSL paradigms is contrastive learning, which learns representations by estimating and maximizing the mutual information between the input and the output of a deep neural network encoder [8].

For graphs, some recent graph contrastive learning methods [29, 7, 41, 25, 20, 19, 43, 36, 40] have shown promising competitiveness compared with semi-supervised GNNs. They usually require three typical steps: (1) constructing positive examples (semantically correlated structural instances) by strategies such as node dropping, edge perturbation, and negative examples (uncorrelated instances) by strategies such as feature shuffling, mini-batch sampling; (2) encoding these examples through graph encoders such as GCN [16]; (3) maximizing/minimizing the similarity between these positive/negative

---

\*Corresponding author

36th Conference on Neural Information Processing Systems (NeurIPS 2022).

examples. Nevertheless, in the real world, graphs often contain multiple types of objects and multiple types of relationships between them, which are called heterogeneous graphs, or heterogeneous information networks (HINs) [27]. Due to the challenges caused by the heterogeneity, existing SSL methods on homogeneous graphs cannot be straightforwardly applied to HINs. Very recently, several works have made some efforts to conduct SSL on HINs [33, 23, 18, 15, 13, 14, 37, 10]. In comparison with SSL methods on homogeneous graphs, the key difference is that they usually have different example generation strategies, so as to capture the heterogeneous structural properties in HINs.

The strategies of generating high-quality positive/negative examples are critical to the performance of existing methods [34, 4, 41, 36]. Unfortunately, whether for homogeneous graphs or heterogeneous graphs, the example generation strategies are dataset-specific, and may not be applicable to all scenarios. This is because real-world graphs are abstractions of things from various domains, e.g., social networks, citation networks, etc. They usually have significantly different structural properties and semantics. Previous works have systematically studied this and found that different strategies are good at capturing different structural semantics. For example, study [41] observed that edge perturbation benefits social networks but hurts some biochemical networks, and study [36] observed that negative examples benefit sparser graphs. Consequently, in practice, the example generation strategies have to be empirically constructed and investigated through either trial-and-error or rules of thumb. This significantly limits the practicality and general applicability of existing methods.

In this work, we focus on HINs which are more challenging, and propose a novel SSL approach, named SHGP. Different from existing methods, *SHGP requires neither positive examples nor negative examples*, thus circumventing the above issues. Specifically, SHGP adopts any HGNN model that is based on attention-aggregation scheme as the base encoder, which is termed as the module Att-HGNN. The attention coefficients in Att-HGNN are particularly used to combine with the structural clustering method LPA (label propagation algorithm) [21], as the module Att-LPA. Through performing structural clustering on HINs, Att-LPA is able to produce clustering labels, which are treated as pseudo-labels. In turn, these pseudo-labels serve as guidance signals to help Att-HGNN learn better embeddings as well as better attention coefficients. Thus, the two modules are able to exploit and enhance each other, finally leading the model to learn discriminative and informative embeddings. In summary, we have three main contributions as follows:

- We propose a novel SSL method on HINs, SHGP. It innovatively consists of the Att-LPA module and the Att-HGNN module. The two modules can effectively enhance each other, facilitating the model to learn effective embeddings.

- To the best of our knowledge, SHGP is the first attempt to perform SSL on HINs without any positive or negative examples. Therefore, it can directly avoid the laborious investigation of example generation strategies, improving the model's generalization ability and flexibility.

- We transfer the object embeddings learned by SHGP to various downstream tasks. The experimental results show that SHGP can outperform state-of-the-art baselines, even including some semi-supervised baselines, demonstrating its superior effectiveness.

## 2   Related work

**SSL on HINs.**   There are several existing methods [33, 23, 18, 15, 37, 13, 14, 10] that conduct SSL on HINs. Determined by their contrastive loss functions, all these methods require high-quality positive and negative examples to effectively learn embeddings. Thus, their effectiveness and performance hinge on the specific strategies of generating positive examples and negative examples, which limits their flexibility and generalization ability.

**SSL on homogeneous graphs.**   Existing SSL methods on homogeneous graphs [29, 7, 41, 25, 20, 19, 43, 36, 35, 30] also need to generate sufficient positive and negative examples to effectively perform graph contrastive learning. They only handle homogeneous graphs and cannot be easily applied to HINs. In this work, we seek to perform SSL on HINs without any positive examples or negative examples.

**GNN+LPA methods.**   There exist several methods [31, 1, 24] that combine LPA [21] with GNNs. However, they are all supervised learning methods, and only deal with homogeneous graphs. In this work, we study SSL on HINs.

**Others.** DeepCluster [2] uses $K$-means to perform clustering in the embedding space. Differently, our SHGP directly performs structural clustering in the graph space. JOAO [40] explores the automatic selection of positive example generation strategies, which is still not fully automatic. HuBERT [9] is an SSL approach for speech representation learning. GIANT [3] leverages graph-structured self-supervision to extract numerical node features from raw data. MARU [12] learns object embeddings by exploiting meta-contexts in random walks. Different from them, in this work, we study how to effectively conduct SSL on HINs. Graph pooling methods e.g. [22, 39] learn soft cluster assignment to coarsen graph topology in each model layer. Differently, our method propagates integer (hard) cluster labels in each layer to perform structural clustering.

## 3  Preliminaries

We first briefly introduce some concepts about HINs, and then formally describe the problem we study in this paper.

**Heterogeneous Information Network.** An HIN is defined as: $\mathcal{G} = (\mathcal{V}, \mathcal{E}, \mathcal{A}, \mathcal{R}, \phi, \psi)$, where $\mathcal{V}$ is the set of objects, $\mathcal{E}$ is the set of links, $\phi : \mathcal{V} \to \mathcal{A}$ and $\psi : \mathcal{E} \to \mathcal{R}$ are respectively the object type mapping function and the link type mapping function, $\mathcal{A}$ denotes the set of object types, and $\mathcal{R}$ denotes the set of relations (link types), where $|\mathcal{A}| + |\mathcal{R}| > 2$. Let $\mathcal{X} = \{\mathbf{X}_1, ..., \mathbf{X}_{|\mathcal{A}|}\}$ denote the set containing all the feature matrices associated with each type of objects. A meta-path $\mathcal{P}$ of length $l$ is defined in the form of $A_1 \xrightarrow{R_1} A_2 \xrightarrow{R_2} \cdots \xrightarrow{R_l} A_{l+1}$ (abbreviated as $A_1 A_2 \cdots A_{l+1}$), which describes a composite relation $R = R_1 \circ R_2 \circ \cdots \circ R_l$ between object types $A_1$ and $A_{l+1}$, where $\circ$ denotes the composition operator on relations.

We show a toy HIN in the left part of Figure 1. It contains four object types: "Paper" ($P$), "Author" ($A$), "Conference" ($C$) and "Term" ($T$), and three relations: "Publish" between $P$ and $C$, "Write" between $P$ and $A$, and "Contain" between $P$ and $T$. $APC$ is a meta-path of length two, and $a_1 p_2 c_2$ is such a path instance, which means that author $a_1$ has published paper $p_2$ in conference $c_2$.

**SSL on HINs.** Given an HIN $\mathcal{G}$, the problem is to learn an embedding vector $\mathbf{h}_i \in \mathbb{R}^d$ for each object $i \in \mathcal{V}$, in a self-supervised manner, i.e., without using any task-specific labels. The pre-trained embeddings are expected to capture the general-purpose information contained in $\mathcal{G}$, and can be easily transferred to various unseen downstream tasks with only a few task-specific labels.

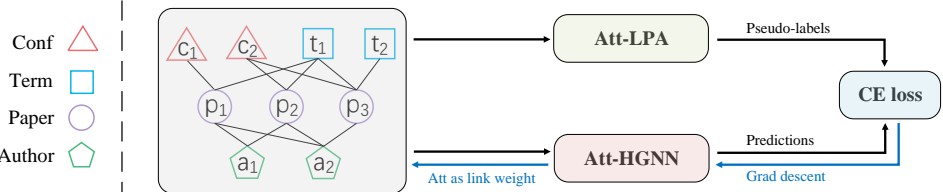

Figure 1: The overall architecture of SHGP. Given an HIN, in each iteration, we use Att-HGNN to produce embeddings and predictions, and use Att-LPA to produce pseudo-labels. The loss is computed as the cross-entropy between the predictions and the pseudo-labels. The attention coefficients (and other parameters) are optimized via gradient descent, which serve as the new attention-aggregation weights of Att-HGCN and Att-LPA in the next iteration, promoting them to produce better embeddings and predictions, as well as better pseudo-labels.

## 4  Methodology

In this section, we present the proposed method SHGP, which consists of two key modules. The Att-HGNN module is instantiated as any HGNN model that is based on attention-aggregation scheme. The Att-LPA module combines the structural clustering method LPA [21] with the attention-aggregation scheme used in Att-HGNN. The overall model architecture is shown in Figure 1 and explained in the figure caption. In the following, we describe the procedure of SHGP in detail.

## 4.1 Initialization

At the beginning, we randomly initialize all the model parameters in Att-HGNN by the Xavier uniform distribution [6]. To obtain initial pseudo-labels, we use the original LPA [21] to perform a thorough structural clustering on the input HIN $\mathcal{G}$. Specifically, LPA randomly associates each object with a unique integer as its initial label, and lets them iteratively propagate along the links in $\mathcal{G}$. In each iteration, each object updates its label to the label that appears most frequently in its neighborhood. After convergence, the final label indicates the cluster to which each object belongs. We treat these clustering labels returned by LPA as the initial pseudo-labels, and re-organize them as a one-hot label matrix $\mathbf{Y}^{[0]} \in \mathbb{R}^{|\mathcal{V}| \times K}$, where $K$ denotes the cluster size, which is not a hyper-parameter but depends on the uniqueness of these labels. Thus, in the subsequent steps, the propagation of the pseudo-labels can be easily implemented via the matrix multiplication operation.

Under the guidance of these obtained initial pseudo-labels, we first train Att-HGNN for several epochs as the model "warm-up", to learn the initial meaningful attention coefficients as well as other model parameters. We use $\mathcal{W}^{[0]}$ to denote all these initial parameters. Then, we proceed with the following iterations.

## 4.2 Iteration

In the $t$-th iteration, we compute the object embeddings $\mathbf{H}^{[t]}$ by the Att-HGNN module. Att-HGNN is parameterized by $\mathcal{W}^{[t-1]}$, and its inputs include: the HIN topology $\mathcal{G}$, the object features $\mathcal{X}$. This is formulated as follows:

$$\mathbf{H}^{[t]} = \textbf{Att-HGNN}\big(\mathcal{W}^{[t-1]}, \mathcal{G}, \mathcal{X}\big) \tag{1}$$

Meanwhile, we update the pseudo-labels by the creatively proposed Att-LPA module. In comparison with Att-HGNN, Att-LPA does not input $\mathcal{X}$, but instead inputs $\mathbf{Y}^{[t-1]}$, i.e., the pseudo-labels of the previous iteration. This is formulated as follows:

$$\mathbf{Y}^{[t]} = \textbf{Att-LPA}\big(\mathcal{W}^{[t-1]}, \mathcal{G}, \mathbf{Y}^{[t-1]}\big) \tag{2}$$

Att-HGNN and Att-LPA perform one forward pass of attention-based aggregation in the same way. The only difference between them is that Att-HGNN aggregates the (projected) features of neighbors while Att-LPA aggregates the pseudo-labels of neighbors produced in the previous iteration, both weighted by exactly the same attention coefficients.

Now, we input $\mathbf{H}^{[t]}$ into a softmax classifier which is built on the top layer of the Att-HGNN to make predictions $\mathbf{P}^{[t]}$. The loss is computed as the cross-entropy between the predictions $\mathbf{P}^{[t]}$ and the pseudo-labels $\mathbf{Y}^{[t]}$, as follows:

$$\mathbf{P}^{[t]} = softmax(\mathbf{H}^{[t]} \cdot \mathbf{C}^{[t-1]})$$
$$\mathcal{L}^{[t]} = -\sum_{i \in \mathcal{V}} \sum_{c=1}^{K} \mathbf{Y}^{[t]}_{i,c} \ln \mathbf{P}^{[t]}_{i,c} \tag{3}$$

where $\mathbf{C}^{[t-1]} \subset \mathcal{W}^{[t-1]}$ denotes the parameter matrix of the classifier.

Having the loss, we can then optimize all the model parameters by gradient descent, as follows:

$$\mathcal{W}^{[t]} = \mathcal{W}^{[t-1]} - \eta \cdot \nabla_{\mathcal{W}} \mathcal{L}^{[t]} \tag{4}$$

where $\eta$ denotes the learning rate. As the optimization proceeds, the model will learn better model parameters (including attention coefficients). This leads Att-HGNN and Att-LPA to respectively produce better embeddings (and predictions) and better pseudo-labels in the next iteration, which, in turn, promotes the model to learn further better parameters. Thus, the two processes are able to closely interact with each other, as well as enhance each other, finally resulting in discriminative and informative embeddings. The overall procedure is shown in Algorithm 1.

## 4.3 Model discussion

### 4.3.1 Encoder of Att-HGNN

The Att-HGNN module can be specifically instantiated as any attention-based HGNN encoders. Existing possible choices include: HAN [32], HGT [11], GTN [42], and ie-HGCN [38]. Among them,

---
**Algorithm 1** The overall procedure of SHGP
---
**Input**: An HIN $\mathcal{G}$
**Output**: Object embeddings
 1: Perform a thorough LPA process to get initial pseudo-labels.
 2: Guided by the initial pseudo-labels, warm-up Att-HGNN for several epochs.
 3: **while** Not Converged **do**
 4:  Perform one forward pass of Att-HGNN module to compute embeddings by Eq. (1).
 5:  Perform one forward pass of Att-LPA module to update pseudo-labels by Eq. (2).
 6:  Compute cross-entropy loss by Eq. (3).
 7:  Optimize all the model parameters (including attention coefficients) by Eq. (4).
 8: **end while**
---

our previously proposed ie-HGCN [38] is simple and efficient. It has shown superior performance over the other three models. Therefore, in this work, we adopt it as the base encoder of Att-HGNN, which is in detail described as follows.

Let $\mathcal{N}_i$ denote the set of object $i$'s neighbors. Specially, object $i$ itself is also added to $\mathcal{N}_i$. Accordingly, an edge $(i, i)$ is added to $\mathcal{E}$, and a dummy self-relation $\psi(i, i)$ is added to $\mathcal{R}$. In each model layer, object $i$'s new representation $\vec{h}_i'$ is computed as follows:

$$\vec{h}_i' = \sigma\left( \sum_{j \in \mathcal{N}_i} \beta_i^{\psi(i,j)} \cdot \widehat{a}_{i,j}^{\psi(i,j)} \cdot \mathbf{W}^{\psi(i,j)} \cdot \vec{h}_j \right) \tag{5}$$

where $\sigma$ is the non-linear activation function, and $\vec{h}_j$ is neighbor $j$'s current representation. $\mathbf{W}^{\psi(i,j)}$ is the projection parameter matrix, $\widehat{a}_{i,j}^{\psi(i,j)}$ is the normalized link weight ($\widehat{a}_{i,i}^{\psi(i,i)} = 1$), and $\beta_i^{\psi(i,j)}$ is the normalized attention coefficient, all of which are specific to relation $\psi(i, j)$.

With Att-HGNN, we can formulate our Att-LPA in a similar way. Specifically, in each layer, object $i$'s pseudo-label is updated as follows:

$$\vec{y}_i\prime = \textit{one-hot}\left( \textit{argmax}\left( \sum_{j \in \mathcal{N}_i} \beta_i^{\psi(i,j)} \cdot \widehat{a}_{i,j}^{\psi(i,j)} \cdot \vec{y}_j \right) \right) \tag{6}$$

where $\vec{y}_j$ is neighbor $j$'s pseudo-label vector in the one-hot form. Object $i$ first aggregates its neighbors' pseudo-labels according to the same normalized link weights and attention coefficients as in Eq. (5). Then, its new pseudo-label vector $\vec{y}_i\prime$ is obtained through the argmax operator and the one-hot encoding function.

The above two equations describe the computational details of one model layer, where the iteration indices and the layer indices are omitted for notation brevity. Here, we can further use the superscript $[x, y]$ to index a symbol with respect to $x$-th iteration and $y$-th layer. Assume the model has $N$ layers. In the $t$-th iteration, for the input layer of Att-HGNN, we set: $\vec{h}_i^{[t,0]} = \vec{x}_i$ (i.e., the object feature vector), while for Att-LPA, we set: $\vec{y}_i^{[t,0]} = \vec{y}_i^{[t-1,N]}$. At the end of the $t$-th iteration, we properly organize all the $\vec{h}_i^{[t,N]}$ and $\vec{y}_i^{[t,N]}$ as $\mathbf{H}^{[t]}$ in Eq. (1) and $\mathbf{Y}^{[t]}$ in Eq. (2) respectively. In this way, based on the ie-HGCN encoder, we establish our Att-HGNN module and Att-LPA module.

### 4.3.2 Time complexity

As analyzed in the literature, both LPA [21] and ie-HGCN [38] have quasi-linear time complexity. Therefore, our SHGP also has quasi-linear time complexity, i.e., $\mathcal{O}(|\mathcal{V}| + |\mathcal{E}|)$. Please see Appendix A.2 for the efficiency study of SHGP.

### 4.3.3 Label consistency

For SHGP, in the initialization phase, we perform the thorough LPA process until convergence to obtain the initial pseudo-labels, and warm-up Att-HGNN module to learn initial meaningful attention coefficients. In each subsequent iteration, we no longer need to perform LPA from scratch, but instead perform one forward pass of Att-LPA. Therefore, the clustering results are basically consistent

with the results of the previous iteration. Thus, our SHGP does not need to perform alignment between clustering labels and predictions. This can avoid the issue in DeepCluster [2] that there is no correspondence between two consecutive clustering assignments, and can also avoid the alignment process in M3S [26].

### 4.3.4 Class space

In this work, we cluster all types of objects into one common class space. We note that in most existing HIN datasets, various types of objects can indeed share the same class space. They typically show a "star" network schema [28], since they are usually constructed according to one "hub" type of objects. The other types of objects are actually the attributes of the "hub" objects. For example, on DBLP, the "hub" objects are paper ($P$) objects. These papers and their associated authors, conferences, and etc. can all be categorized into four research areas: DM, IR, DB, and AI.

In the following extensive experiments, we demonstrate that this strategy works well on existing widely used HIN benchmark datasets. In future work, we will further investigate the problem of clustering different types of objects into different class spaces.

## 5 Experiments

In this section, we verify the generalization ability of the proposed SHGP by transferring the pre-trained object embeddings to various downstream tasks including object classification, object clustering, and embedding visualization.

### 5.1 Datasets

In the experiments, we use four publicly available HIN benchmark datasets, which are widely used in previous related works [38, 32, 18, 23, 33]. Their statistics are summarized in Table 1. Please see Appendix A.1 for more details of these datasets.

Table 1: Dataset statistics.

| Datasets | Objects (number) | Relations |
|----------|------------------|-----------|
| MAG | $P$ (4017), $A$ (15383), $I$ (1480), $F$ (5454) | $P \rightleftharpoons P, P \rightleftharpoons F, P \rightleftharpoons A, A \rightleftharpoons I$ |
| ACM | $P$ (4025), $A$ (7167), $S$ (60) | $P \rightleftharpoons A, P \rightleftharpoons S$ |
| DBLP | $A$ (4057), $P$ (14328), $C$ (20), $T$ (8898) | $P \rightleftharpoons A, P \rightleftharpoons C, P \rightleftharpoons T$ |
| IMDB | $M$ (3328), $A$ (42553), $U$ (2103), $D$ (2016) | $M \rightleftharpoons A, M \rightleftharpoons U, M \rightleftharpoons D$ |

- **MAG** is a subset of Microsoft Academic Graph. It contains four object types: Paper ($P$), Author ($A$), Institution ($I$) and Field ($F$), and eight relations between them. Paper objects are labeled as four classes according to their published venues: IEEE Journal of Photovoltaics, Astrophysics, Low Temperature Physics, and Journal of Applied Meteorology and Climatology.

- **ACM** is extracted from ACM digital library. It contains three object types: Paper ($P$), Author ($A$) and Subject ($S$), and four relations between them. Paper objects are divided into three classes: Data Mining, Database, and Computer Network.

- **DBLP** is extracted from DBLP bibliography. It contains four object types: Author ($A$), Paper ($P$), Conference ($C$) and Term ($T$), and six relations between them. Author ($A$) objects are labeled according to their four research areas: Data Mining, Information Retrieval, Database, and Artificial Intelligence.

- **IMDB** is extracted from the online movie rating website IMDB. It contains four object types: Movie ($M$), Actor ($A$), User ($U$) and Director ($D$), and six relations between them. Movie ($M$) objects are categorized into four classes according to their genres: Comedy, Documentary, Drama, and Horror.

### 5.2 Baselines

We compare our SHGP with seven HIN-oriented baselines, including two semi-supervised methods, and five unsupervised (self-supervised) methods:

- **Semi-supervised**: HAN [32], and ie-HGCN (abbreviated as HGCN) [38].
- **Unsupervised**: metapath2vec (abbreviated as M2V) [5], DMGI [18], HDGI [23], HeCo [33], and HGCN-DC (abbreviated as H-DC).

Here, HAN and HGCN are semi-supervised HGNN methods. M2V is a traditional unsupervised HIN embedding method. DMGI, HDGI, and HeCo are state-of-the-art SSL methods on HINs. Note that, in the experiments, we adopt the baseline HGCN's encoder as the base encoder of our Att-HGNN. H-DC is a variant of our SHGP. It also uses HGCN as the base encoder, but unlike SHGP which performs structural clustering in the graph space, H-DC uses $K$-means to perform clustering in the embedding space, like DeepCluster [2]. We can use baselines HGCN and H-DC to investigate the effectiveness of the proposed self-supervised pre-training scheme based on structural clustering.

### 5.3 Implementation details

For the proposed SHGP, in all the experiments, we use two HGCN layers as the Att-HGNN encoder, and search the dimensionalities of the hidden layers in the set {64, 128, 256, 512}. All the model parameters are initialized by the Xavier uniform distribution [6], and they are optimized through the Adam optimizer. The learning rate and weight decay are searched from 1e-4 to 1e-2. For the number of warm-up epochs, we search its best value in the set {5, 10, 20, 30, 40, 50}. We pre-train SHGP with up to 100 epochs and select the model with the lowest validation loss as the pre-trained model. Then, we freeze the model and transfer the learned object embeddings to various downstream tasks. For baselines, we reproduce their experimental results on our datasets. Their hyper-parameters are searched based on their papers and their official source codes. Some of the baseline methods [32, 5, 18, 33] need users to input several meta-paths, which are specified in Appendix A.1 in detail. For all the methods, we randomly repeat all the evaluation tasks for ten times and report the average results. All the experiments are conducted on an NVIDIA GTX 1080Ti GPU.

Table 2: Object classification results (%).

| Datasets | Metrics | Train | HAN | HGCN | M2V | DMGI | HDGI | HeCo | H-DC | SHGP |
|---|---|---|---|---|---|---|---|---|---|---|
| MAG | Mic-F1 | 4% | 90.07 | 93.16 | 88.97 | 94.43 | 94.10 | 95.75 | 85.03 | **98.23** |
| | | 6% | 91.83 | 95.18 | 89.94 | 93.80 | 93.68 | 95.93 | 85.16 | **98.30** |
| | | 8% | 92.17 | 97.13 | 90.15 | 94.36 | 94.27 | 96.08 | 86.03 | **98.37** |
| | Mac-F1 | 4% | 89.93 | 92.82 | 88.51 | 94.32 | 93.89 | 95.27 | 84.72 | **98.24** |
| | | 6% | 91.54 | 95.08 | 89.45 | 93.74 | 93.64 | 95.42 | 85.13 | **98.33** |
| | | 8% | 91.82 | 97.05 | 89.73 | 94.27 | 94.23 | 95.15 | 85.97 | **98.41** |
| ACM | Mic-F1 | 4% | 70.84 | 75.78 | 72.45 | 78.93 | 79.72 | 79.78 | 78.53 | **80.31** |
| | | 6% | 72.04 | 77.59 | 73.83 | 79.01 | 80.09 | 80.15 | 79.96 | **80.78** |
| | | 8% | 73.23 | 78.08 | 73.95 | 79.47 | 79.07 | **80.94** | 79.82 | 80.91 |
| | Mac-F1 | 4% | 61.50 | 64.61 | 53.01 | 59.37 | 60.57 | 65.91 | 64.89 | **67.14** |
| | | 6% | 60.23 | 64.04 | 51.86 | 59.15 | 61.09 | 65.63 | 64.37 | **67.38** |
| | | 8% | 62.37 | 65.73 | 53.72 | 59.42 | 59.99 | 67.15 | 65.11 | **68.19** |
| DBLP | Mic-F1 | 4% | 90.48 | 92.45 | 88.93 | 89.35 | 88.33 | 91.31 | 87.15 | **93.70** |
| | | 6% | 91.03 | 92.08 | 89.47 | 89.21 | 88.93 | 91.05 | 86.67 | **93.92** |
| | | 8% | 91.90 | 92.34 | 91.41 | 89.88 | 88.18 | 91.22 | 87.23 | **94.13** |
| | Mac-F1 | 4% | 90.01 | 92.13 | 88.49 | 88.21 | 87.69 | 90.53 | 87.03 | **93.31** |
| | | 6% | 90.51 | 91.71 | 88.97 | 88.03 | 88.75 | 90.26 | 86.53 | **93.52** |
| | | 8% | 91.35 | 92.04 | 89.83 | 88.57 | 87.38 | 90.42 | 87.11 | **93.77** |
| IMDB | Mic-F1 | 4% | 56.05 | 56.68 | 56.54 | 54.79 | 56.31 | 57.42 | 54.01 | **58.51** |
| | | 6% | 54.21 | 57.72 | 55.24 | 54.93 | 57.64 | 58.63 | 54.19 | **59.76** |
| | | 8% | 56.45 | 57.03 | 57.02 | 55.75 | 56.70 | 60.13 | 55.19 | **61.60** |
| | Mac-F1 | 4% | 39.04 | 36.66 | 27.03 | 37.95 | 30.84 | 38.66 | 34.72 | **43.36** |
| | | 6% | 36.63 | 39.38 | 26.51 | 38.67 | 36.35 | 39.43 | 36.61 | **46.17** |
| | | 8% | 38.20 | 40.54 | 27.86 | 39.89 | 34.64 | 40.00 | 38.03 | **48.02** |

## 5.4 Object classification

We conduct object classification on the object embeddings learned by all the semi-supervised baselines and unsupervised baselines. On each dataset, for the objects that have ground-truth labels, we randomly select $\{4\%, 6\%, 8\%\}$ objects as the training set. The others are divided equally as the validation set and the test set. For all the unsupervised methods, they don't use any class labels during learning object embeddings. After finishing the pre-training, the output object embeddings and their corresponding labels are used to train a linear logistic regression classifier. For all the semi-supervised methods, we directly report the classification results output by their own classifiers. We adopt Micro-F1 and Macro-F1 as evaluation metrics. The results are reported in Table 2.

We can see that our proposed SHGP achieves the best overall performance, even exceeding several semi-supervised learning methods, indicating its superior effectiveness. On MAG, SHGP achieves very high performance, i.e., over 98% Micro-F1/Macro-F1 scores, which are close to saturation. Recall that SHGP adopts HGCN as the base encoder, and here, SHGP achieves better performance than HGCN. This indicates the effectiveness of the proposed strategy, i.e., pre-training HGNNs in a self-supervised manner based on structural clustering. SHGP also performs better than H-DC. This is reasonable because H-DC uses $K$-means to perform clustering in the embedding space to obtain pseudo-labels, and thus it cannot effectively exploit the structural information which naturally resides in the graph space.

Most unsupervised pre-training methods outperform traditional semi-supervised methods. This verifies the superiority of the recent advances in the self-supervised pre-training paradigm. Among the unsupervised methods, M2V performs much worse than the others. This is because M2V can only exploit a single meta-path, unlike DMGI, HDGI and HeCo which can well fuse the information conveyed by multiple meta-paths through regularization or attention mechanism. Among the semi-supervised methods, HGCN outperforms HAN, probably because HAN can only exploit user-specified meta-paths while HGCN can automatically discover and exploit the most useful meta-paths [38].

## 5.5 Object clustering

We evaluate the embeddings through object clustering. In this task, we only consider the unsupervised methods, and the semi-supervised methods are excluded because they have exploited class labels during learning embeddings. On each dataset, we use $K$-means to cluster the embeddings of the labeled objects, and report normalized mutual information (NMI) and adjusted rand index (ARI) to quantitatively assess the clustering quality.

Table 3: Object clustering results (%).

|  | MAG | | ACM | | DBLP | | IMDB | |
| --- | --- | --- | --- | --- | --- | --- | --- | --- |
|  | NMI | ARI | NMI | ARI | NMI | ARI | NMI | ARI |
| M2V | 39.67 | 43.75 | 32.53 | 28.49 | 49.50 | 56.73 | 1.43 | 1.03 |
| DMGI | 70.89 | 73.51 | 38.45 | 32.46 | 65.17 | 67.23 | 3.49 | 2.65 |
| HDGI | 73.96 | 77.15 | 39.13 | 32.34 | 59.98 | 62.33 | 4.15 | 2.96 |
| HeCo | 79.33 | 83.16 | 39.06 | **32.69** | 68.81 | 74.05 | 5.69 | 2.32 |
| H-DC | 42.75 | 49.01 | 18.60 | 19.75 | 47.15 | 53.15 | 1.57 | 1.12 |
| SHGP | **90.65** | **93.00** | **39.42** | 32.63 | **73.30** | **77.31** | **6.33** | **3.10** |

As shown in Table 3, our SHGP achieves the best overall performance in this task. Especially, on MAG, SHGP outperforms other baselines by a large margin. This demonstrates its superior competitiveness against other baselines in terms of learning discriminative embeddings. On IMDB, all the methods have shown limited performance. This may be because the class labels and the structural information on this dataset are not significantly correlated. M2V shows the worst performance again, which may be due to its limitation as we analyzed in the previous experiment.

## 5.6 Visualization

For a more intuitive comparison, on MAG, we visualize the paper objects' embeddings pre-trained by three unsupervised methods, i.e., DMGI, HeCo, and our SHGP. The embeddings are further

embedded into the 2-dimensional Euclidean space by the t-SNE algorithm [17], and they are colored according to their ground-truth labels.

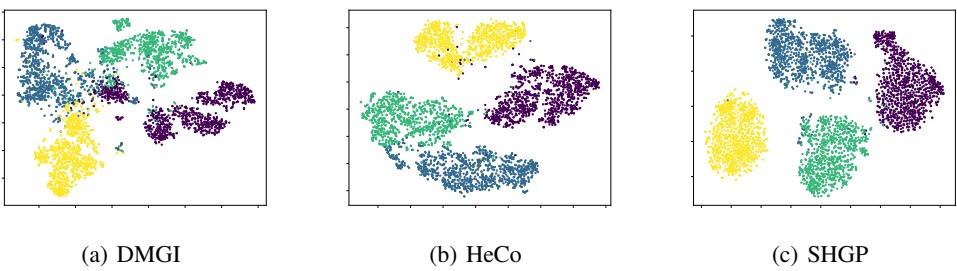

(a) DMGI            (b) HeCo            (c) SHGP

Figure 2: Visualization of the pre-trained embeddings of paper objects on MAG.

The results are shown in Figure 2. We can see that DMGI shows blurred boundaries between different classes. HeCo performs relatively better than DMGI, but the green (left) class objects and the blue (bottom) class objects are still mixed to some extent. Our SHGP shows the best within-class compactness and the clearest between-class boundaries, which demonstrates its superior effectiveness.

## 5.7  Hyper-parameter study

In this subsection, we investigate the sensitivity of the warm-up epochs, which is the main hyper-parameter that we have introduced in Section 4.1. Specifically, on all the datasets, we explore the change of object classification performance as the number of warm-up epochs gradually increases. The results are shown in Figure 3.

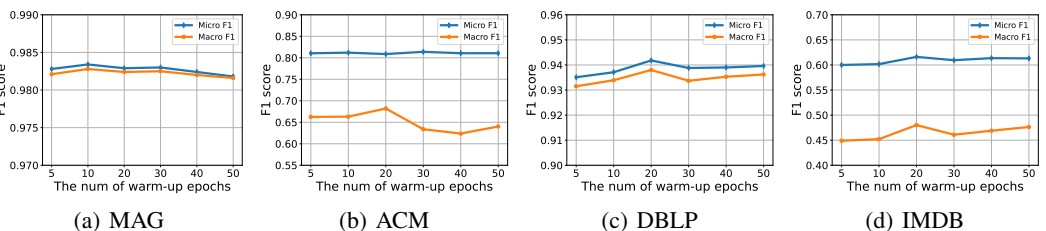

(a) MAG        (b) ACM        (c) DBLP        (d) IMDB

Figure 3: Analysis of the number of warm-up epochs.

As shown, overall, this hyper-parameter is not very sensitive. The general pattern is that the performance increases first, and then declines gradually. This is because, at the beginning, the model has not made full use of the information contained in the initial pseudo-labels yet. After that, the model may gradually overfit these initial pseudo-labels, which prevents the model from continuously improving its performance in the iterations. The overall inflection points are 10, 20, 20, and 20 for MAG, ACM, DBLP, and IMDB respectively, which are the default values in the other experiments.

## 6  Conclusion

In this paper, we propose a novel self-supervised pre-training method on HINs, named SHGP. It consists of two key modules which share the same attention-aggregation scheme. The two modules are able to utilize and enhance each other, promoting the model to effectively learn informative embeddings. Different from existing SSL methods on HINs, our SHGP does not require any positive examples or negative examples, thereby enjoying a high degree of flexibility and ease of use. We conduct extensive experiments to demonstrate the superior effectiveness of SHGP against state-of-the-art baselines.

## Societal impact

Heterogeneous Information Networks (HINs) widely exist in our society, such as social networks, power networks, virus networks, etc. In this work, we develop a method to learn the representations for objects in HINs. The learned representations can be used for various analytical tasks such as object classification, object clustering, link prediction, etc. We believe that our method contributes to our society in many aspects. However, there is also some risk that our method could be abused illegally or unethically for some undesirable purposes. We hope those bad things would not happen.

## Funding transparency statement

This work was supported by the National Natural Science Foundation of China under Grants 62133012, 61936006, 62103314, 62203354, 61876144, 61876145, 62073255, 61876138 and 62002255, the Key Research and Development Program of Shaanxi under Grant 2020ZDLGY04-07, and the Innovation Capability Support Program of Shaanxi under Grant 2021TD-05.

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
