# OpenReview forum: "Self-supervised Heterogeneous Graph Pre-training Based on Structural Clustering"
_NeurIPS.cc/2022/Conference — NeurIPS 2022 Accept_

### Official Review · Reviewer_eMAZ · 2022-07-10

**Rating:** 5
**Confidence:** 3
**Soundness:** 2 fair
**Presentation:** 3 good
**Contribution:** 2 fair

**Summary:**

This paper studies self-supervised learning (SSL) techniques for heterogeneous graph neural networks (HGNN). The proposed SSL method, SHGP, optimized attention-based HGNN layer (Attn-HGNN) by cross-entropy loss between encoder’s predictions and node’s pseudo cluster label obtained from the label propagation algorithm (Attn-LPA). In addition, SHGP demonstrated that iteratively training model parameters based on the updated pseudo labels from Attn-LPA can further improve the model performance.

**Questions:**

1. It is unclear which part of the structural clustering leads to the final good performance, as there are several hyper-parameters in pseudo node label cluster generation. For example,
- How performance would change if we freeze (or only update only very few times) the pseudo node label, but just optimize the attn-HGCN layer?
- How does the number of cluster sizes (K) affect the embedding quality? Can we also dynamically adjust the cluster size based on some coarse-to-fine heuristic?
- If each node comes with side information (e.g., text description), can we cluster nodes based on label text representation?
How does the pseudo label assignment change between two attn-LPA updates? For example, how many nodes change their pseudo label cluster?

2. Some implementation and model details are missing in Sec 5.3 and Table 1. For example,
- Comparison model parameters between SHGP and baselines, which showed if the performance gain is due to larger encoder size and node embedding dimensions.
- The pre-training time of SHGP and the baselines.


**Limitations:**

No potential negative societal impact.

**Strengths And Weaknesses:**

## Strength
- The paper writing is clear and easy to follow
- The empirical performance seems promising

## Weakness
- SHGP introduces several hyper-parameters, including the number of iterations for warm-up (line 2 in Algorithm 1), the number of iterations for main update (line 3 of Algorithm 1), the number of cluster size K.
- Lack of comprehensive ablation study on structural clustering. See Questions section  for further comments.
- Lack of proper citation for SSL based on dynamically-updated pseudo labels in the literature. For example, HuBERT (Hsu et al.) also considers pre-training tasks based on  pseudo labels, where k-mean clusters are dynamically updated.

---

> ### Author Response · Authors · 2022-08-02
> **Response to Reviewer eMAZ (Part 2)**
>
> **Point 5**: *If each node comes with side information (e.g., text description), can we cluster nodes?*
>
> **Reply 5**: At present, we cluster nodes by the Att-LPA module without consideration of the side information. Nevertheless, the Att-HGNN module is able to effectively exploit side information. As we explained in the paper, the two modules are able to closely interact with and enhance each other in each iteration. Therefore, the Att-LPA module and its generated pseudo-labels can exploit side information in an indirect way. It turns out that our current solution works well on widely used datasets. In future work, we would like to explore upgrading the Att-LPA module to directly exploit side information.
>
> ---
>
> **Point 6**: *Lack of proper citation for SSL based on dynamically-updated pseudo labels, e.g., HuBERT (Hsu et al.).*
>
> **Reply 6**: We guess you are referring to the literature *[TASLP 2021] HuBERT: Self-Supervised Speech Representation Learning by Masked Prediction of Hidden Units*.  According to your suggestion, we have added the discussion and citation of this literature in our updated manuscript. Please see the last paragraph of Section 2 (Line 92).
>
> ---
>
> **Point 7**: *The pre-training time of SHGP and the baselines are missing.*
>
> **Reply 7**: First, we would like to kindly remind the reviewer that this comment involves one **factual error**. In Section 4.3.2 of the original manuscript, after analyzing the time complexity of SHGP, we referred readers (Lines 188-189) to Appendix A.2 for the efficiency study of SHGP, where we showed the pre-training time of SHGP on eight datasets with different scales. The experimental results verified its quasi-linear time complexity. According to your suggestion, here, we show the pre-training time of several HIN-oriented SSL baselines as well as ours. The total time cost (seconds) consumed by these methods is reported in the table below. As we can see, SHGP is relatively more efficient than these baselines.
>
> *Table 3: Pre-training time cost (seconds).*
>
> | Datasets | DMGI  | HDGI  | HeCo  | SHGP  |
> | :------: | :---: | :---: | :---: | :---: |
> |   MAG    | 11.75 | 16.25 | 39.25 | 10.08 |
> |   ACM    | 10.32 | 14.57 | 26.84 | 9.33  |
> |   DBLP   | 12.93 | 19.44 | 23.28 | 7.41  |
> |   IMDB   | 10.10 | 16.36 | 38.52 | 9.14  |
>
> ---
>
> **Point 8**: *The comparison of model parameters between SHGP and baselines is missing.*
>
> **Reply 8**: Thank you for this comment. According to your suggestion, in the table below, we show the number of model parameters of four HIN-oriented SSL methods. As we can see, compared to these baselines, our model has a reasonable number of parameters.
>
> *Table 4: The number of model parameters.*
>
> | Datasets | DMGI  | HDGI  | HeCo  | SHGP  |
> | :------: | :---: | :---: | :---: | :---: |
> |   MAG    | 2.59M | 0.70M | 2.63M | 2.38M |
> |   ACM    | 2.45M | 0.57M | 1.14M | 0.93M |
> |   DBLP   | 2.53M | 0.74M | 2.59M | 2.40M |
> |   IMDB   | 1.93M | 0.32M | 3.20M | 2.08M |

---

> ### Author Response · Authors · 2022-08-02
> **Response to Reviewer eMAZ (Part 1)**
>
> Thanks for your comments. In the following, we respond to your concerns point by point.
>
> ---
>
> **Point 1**: *SHGP introduces several hyper-parameters, including the number of warm-up iterations, the number of main iterations, and the cluster size K*
>
> **Reply 1**: For the number of warm-up iterations, we studied its sensitivity in Section 5.7. The number of main iterations, i.e., the number of pre-training epochs, can be adaptively determined by various convergence conditions such as pre-training loss (see Section A.4 of the updated Appendix). Moreover, all the baselines involve this hyper-parameter. For the cluster size K, we are sorry for leading you to this confusion. It is not a hyper-parameter but is determined automatically by the LPA algorithm. Please refer to the next response for a detailed explanation. In summary, SHGP only introduces one key hyper-parameter.
>
> ---
>
> **Point 2**: *How does the cluster size K affect the embedding quality? Can we also dynamically adjust it based on some coarse-to-fine heuristic?*
>
> **Reply 2**: The cluster size K here is not a hyper-parameter. It is automatically determined by the LPA. After convergence, LPA returns the cluster label of each object. The cluster size K depends on the uniqueness of these cluster labels. We just encode these cluster labels as one-hot indicator vectors and re-organize them as a label matrix. In our problem, it seems that we have no need to dynamically adjust it based on some coarse-to-fine heuristic. Based on your doubt, we have further clarified the description of K in our updated manuscript. Please see Section 4.1 (Lines 131-132).
>
> ---
>
> **Point 3**: *Lack of comprehensive ablation study on structural clustering.*
>
> **Reply 3**: In the original manuscript, we conducted one ablation study on structural clustering. Specifically, in Section 5.2 (Lines 241-244), we carefully constructed a variant named H-DC which has the same Att-HGNN encoder as SHGP. The difference is that in the variant H-DC, we do not perform structural clustering in the graph space, but use K-means to perform clustering in the embedding space, like DeepCluster. It turns out that SHGP outperforms H-DC in all the cases, which indicates the effectiveness of the structural clustering. Please see the experiments in Section 5.4 and Section 5.5. For more sophisticated ablation studies, we would like to explore them in our future work.
>
> ---
>
> **Point 4**: *How performance would change if we freeze the pseudo node label, but just optimize the attn-HGCN layer?*
>
> **Reply 4**: According to your suggestion, we conduct an ablation study by setting up a variant named SHGP-FPL. In this variant, we first use the original LPA to generate initial pseudo-labels and freeze them. Then, we train the Att-HGNN module under the guidance of these initial pseudo-labels. The quantitative experimental results of object classification are shown in the following table. As we can see, the performance degenerates as we freeze the pseudo labels. This indicates that it is beneficial to update pseudo labels in each iteration.
>
> *Table 1: Micro-F1 scores.*
>
> | Methods  |  MAG  |  ACM  | DBLP  | IMDB  |
> | :------: | :---: | :---: | :---: | :---: |
> | SHGP-FPL | 97.10 | 79.22 | 87.93 | 55.23 |
> |   SHGP   | 98.30 | 80.78 | 93.92 | 59.76 |
>
>
>
> *Table 2: Macro-F1 scores.*
>
> | Methods  |  MAG  |  ACM  | DBLP  | IMDB  |
> | :------: | :---: | :---: | :---: | :---: |
> | SHGP-FPL | 97.15 | 61.18 | 87.71 | 44.80 |
> |   SHGP   | 98.33 | 67.38 | 93.52 | 46.17 |

---

### Official Review · Reviewer_ytZp · 2022-07-11

**Rating:** 6
**Confidence:** 3
**Soundness:** 3 good
**Presentation:** 3 good
**Contribution:** 3 good

**Summary:**

In this paper, a self-supervised heterogeneous pre-training model (SHGP) is proposed. The Att-LPA module obtains node pseudo-labels through structured clustering, and the Att-HGNN module learns node embedding through attention mechanism. SHGP achieved significant results when compared with unsupervised or semi-supervised baseline methods.

**Questions:**

1. Is it necessary to use attention in both ATT-LPA and ATT-HGNN? Ablation study are required.
2. How to overcome the error caused by randomness of LPA algorithm.
3. In Line 5 of algorithm 1, will it be difficult for algorithm to converge if pseudo-labels are continuously updated?


**Limitations:**

Yes.

**Strengths And Weaknesses:**

Strengths:
1. In this paper, a new paradigm of self-supervised pre-training model on heterogeneous graphs is proposed, which generates pseudo-labels of nodes to guide the training of node embedding.
2. In this paper, pseudo-labels are obtained through label propagation, avoiding the generation of positive or negative examples, thus improving the generalization of the model.

Weaknesses:
1. Pseudo-labels initialized by structural clustering only consider structural features and ignore important information in node features.
2. The randomness of pseudo labels generated by LPA algorithm is large.

---

> ### Author Response · Authors · 2022-08-02
> **Response to Reviewer ytZp**
>
> Thanks for your comments. In the following, we respond to your concerns point by point.
>
> ---
>
> **Point 1**: *Is it necessary to use attention in both ATT-LPA and ATT-HGNN? The ablation study is required.*
>
> **Reply 1**: Yes, it is necessary for our method. The shared attention-aggregation scheme between Att-LPA and Att-HGNN is the key factor to bridge the two modules. It facilitates the two modules to continuously enhance each other and improve the model performance. According to your suggestion, we experimentally verify this through an ablation study. We set up a variant named SHGP-NAT by removing the attention aggregation in Att-LPA and Att-HGNN. Specifically, we replace the attention aggregation in Att-HGNN with element-wise mean aggregation, and term it HGNN. Thus, we use the original LPA (no attention) to generate initial pseudo-labels, and train HGNN under the guidance of these initial pseudo-labels. The quantitative experimental results of object classification are shown in the following table. As we can see, the variant SHGP-NAT performs worse than SHGP on all the datasets, which verifies the necessity of the shared attention aggregation in Att-LPA and Att-HGNN.
>
> *Table 1: Micro-F1 scores.*
>
> | Methods  |  MAG  |  ACM  | DBLP  | IMDB  |
> | :------: | :---: | :---: | :---: | :---: |
> | SHGP-NAT | 75.89 | 79.19 | 86.49 | 56.22 |
> |   SHGP   | 98.30 | 80.78 | 93.92 | 59.76 |
>
>
>
> *Table 2: Macro-F1 scores.*
>
> | Methods  |  MAG  |  ACM  | DBLP  | IMDB  |
> | :------: | :---: | :---: | :---: | :---: |
> | SHGP-NAT | 75.74 | 65.90 | 86.27 | 45.07 |
> |   SHGP   | 98.33 | 67.38 | 93.52 | 46.17 |
>
> ---
>
> **Point 2**: *How to overcome the error caused by the randomness of the LPA algorithm.*
>
> **Reply 2**: Indeed, LPA is notorious for its randomness. Nevertheless, in this work, we stick to using LPA to perform structural clustering, because it is efficient and has a similar propagation mechanism as GNNs. In our algorithm, we take two measures to overcome the randomness of LPA: (1) at the very beginning, we execute a thorough LPA process until convergence to obtain reliable initial pseudo-labels; (2) supervised by the obtained initial pseudo-labels, we warm-up Att-HGNN for several epochs to learn reliable initial attention coefficients. The two measures are described in Lines 1-2 of Algorithm 1. After the two initial steps, our algorithms can safely proceed to the subsequent interaction iterations, without being too affected by the randomness.
>
> ---
>
> **Point 3**: *Will it be difficult for the algorithm to converge if pseudo-labels are continuously updated?*
>
> **Reply 3**: In practice, our algorithm converges fast. Specifically, the pre-training of SHGP converges in about 20 iterations on ACM, 30 iterations on DBLP, and 80 iterations on MAG and IMDB. Based on your doubt, we have intuitively demonstrated the convergence by plotting the pre-training loss curves of SHGP. Please see Section A.4 of the updated Appendix. We also provided our source codes in the supplemental material.
>
> ---
>
> **Point 4**: *Pseudo-labels only consider structural features and ignore node features.*
>
> **Reply 4**: The Att-LPA module generates pseudo-labels without consideration of node features. Nevertheless, the Att-HGNN module is able to effectively exploit node features. As we explained in the paper, the two modules are able to closely interact with and enhance each other in each iteration. Therefore, the Att-LPA module and its generated pseudo-labels can exploit node features in an indirect way. It turns out that our current solution works well on widely used datasets. In future work, we would like to explore upgrading the Att-LPA module to directly exploit node features.

---

### Official Review · Reviewer_1dEZ · 2022-07-11

**Rating:** 5
**Confidence:** 4
**Soundness:** 2 fair
**Presentation:** 3 good
**Contribution:** 3 good

**Summary:**

This paper studies the problem of learning object embeddings in heterogeneous information networks. To address this problem, the paper proposes a self-supervised graph learning model SHGP, which consists of two modules, namely Att-HGNN module and Att-LPA module. The main idea of the proposed method is to use the Att-LPA module to generate pseudo labels for objects, which are then leveraged to guide the Att-HGNN module to learn object embeddings. Experimental results on object classification and clustering verify the effectiveness of the method.

**Questions:**

(1) How to choose the initial labels of objects in LPA module? Please see “strength and weakness” for more details about this question.
(2) How to choose the number of clusters $K$ and how does this hyperparameter affects the performance of the proposed method?
(3) Why use LPA to generate pseudo labels instead of other models such as HGNN?

**Limitations:**

The authors do not discuss the limitations and potential negative social impact of the proposed method in the paper.

**Strengths And Weaknesses:**

The idea of using structural clustering (label propagation method) to generate pseudo labels as self-supervision signals is novel and the experimental results verify the effectiveness of such a method. The paper is clear and easy to understand.

However, there are some weaknesses in the paper:

(1) Some technical details of the proposed method are unclear:

a) In section 4.1, the paper writes “LPA randomly associates each object with a unique integer as its initial label”. But where do these “unique integers” come from? Are they chosen from the ground-truth label set?

b) How to determine the number of clusters $K$?

c) In line 171, what is the link weight and how to obtain these weights? Similarly, how to obtain the attention coefficients? Are they model parameters?

(2) The motivation of using LPA to generate pseudo labels is not clear. Why not use another HGNN to generate pseudo labels instead?

(3) In the object classification experiment, I don’t think it is fair to use the object embeddings learned by semi-supervised baselines for comparison as they are not designed to learn object embeddings. Instead, I think the paper should compare the results of unsupervised methods with the results obtained by the classifiers of semi-supervised methods.

Moreover, I am curious about how the number of clusters $K$ affects the performance of SHGP.

Furthermore, the paper claims that the Att-HGNN module can use any attention-based HGNN encoders. However, the performance of the proposed method with different encoders is unknown.

(4) The writing of the paper can be further improved:

a) I doubt whether the sentence in line 21 “This success, however, comes at the cost of a heavy reliance on high-quality supervision labels” is correct since most graph neural network models, such as GCN, GraphSAGE and GAT, are proposed to address the semi-supervised classification problem, which does not require a lot of labels.

b) what do positive examples and negative examples mean?

c) The full name of the proposed model SHGP is missing.

d) I doubt whether the sentence “SHGP is the first attempt to perform SSL on HINs without any positive or negative examples” is correct since the paper does not explicitly explain what is positive examples and negative examples. Can we treat the ground-truth edges and object features as positive examples? If so, then the proposed model still uses positive examples.

e) The citations of datasets are absent in the main paper.

f) minors:
In line 99, conceptions —> concepts;
In line 235, punctuation is missing.

---

> ### Author Response · Authors · 2022-08-02
> **Response to Reviewer 1dEZ (Part 2)**
>
> **Point 7**: *The performance of the proposed method with different encoders is unknown.*
>
> **Reply 7**: Although our Att-HGNN module can be specifically instantiated as various attention-based HGNN encoders, such as HAN, HGT, and GTN, following most previous related works, we directly adopt the most efficient and effective encoder. In this work, we adopt the ie-HGCN encoder since it has quasi-linear time complexity and has shown superior performance over the other three HGNN encoders. Please refer to the original ie-HGCN paper.
>
> ---
>
> **Point 8**: *Why the success of semi-supervised GNNs comes at the cost of a heavy reliance on high-quality supervision labels? Most semi-supervised graph neural network models do not require a lot of labels.*
>
> **Reply 8**: As we know, semi-supervised GNNs learn embeddings under the guidance of supervision labels, and thus their classification performance depends on the quality of class labels for supervision. For graph SSL methods, they leverage the supervision signal from the data itself to learn generalizable embeddings, which can be easily transferred to various downstream tasks with only a few task-specific labels. This advantage has also been verified by multiple previous related studies. e.g., HeCo, DMGI, HDGI, etc.
>
> ---
>
> **Point 9**: *What do positive examples and negative examples mean? Why SHGP is the first attempt to perform SSL on HINs without any positive or negative examples?*
>
> **Reply 9**: In graph-oriented SSL, positive examples are semantically correlated structural instances, which are constructed by graph augmentation strategies such as node dropping, edge perturbation, etc. Negative examples are uncorrelated instances, which are constructed by strategies such as feature shuffling, mini-batch sampling, etc. In this case, our SHGP is the first attempt to perform SSL on HINs without any positive or negative examples. According to your comment, we have updated our manuscript to make it clearer. Please see Paragraph 3 of Section 1 (Lines 33-35).
>
> ---
>
> **Point 10**: *The full name of the proposed model SHGP is missing.*
>
> **Reply 10**: Thank you for this kind reminder. SHGP is the initials of "**S**elf-supervised **H**eterogeneous **G**raph **P**re-training". According to your suggestion, we have marked this by capitalizing the four initials. Please see Abstract (Line 7) in our updated manuscript.
>
> ---
>
> **Point 11**: *The citations of datasets are absent in the main paper.*
>
> **Reply 11**: We would like to kindly remind the reviewer that this comment is a **factual error**. In Section 5.1 (Lines 215-216) of the original manuscript, we cited the datasets and referred readers to Appendix A.1 for more specific details, where we also cited the datasets.
>
> ---
>
> **Point 12**: *There are some minors: in line 99, conceptions —> concepts; in line 235, punctuation is missing.*
>
> **Reply 12**: Thank you for these careful checks. We have revised these minor issues in the updated manuscript.

---

> ### Author Response · Authors · 2022-08-02
> **Response to Reviewer 1dEZ (Part 1)**
>
> Thanks for your comments. In the following, we respond to your concerns point by point.
>
> ---
>
> **Point 1**: *How to choose the initial labels of objects? Are they chosen from the ground-truth label set?*
>
> **Reply 1**: Following the original LPA paper, for a set of objects $\mathcal V$, we initialize the cluster label of each object by randomly sampling an integer from the integer interval [1, $|\mathcal V|$]. They are not chosen from the ground-truth label set.
>
> ---
>
> **Point 2**: *How to determine the number of clusters K? How does it affect the performance?*
>
> **Reply 2**: We are sorry for leading you to this confusion. The number of clusters K here is not a hyper-parameter. It is automatically determined by LPA. After convergence, LPA returns the cluster label of each object. The number of clusters K depends on the uniqueness of these cluster labels. We just encode these cluster labels as one-hot indicator vectors and re-organize them as a label matrix. Based on your doubt, we have further clarified the description of K in our updated manuscript. Please see Section 4.1 (Lines 131-132).
>
> ---
>
> **Point 3**: *Why use LPA to generate pseudo labels instead of other models such as HGNN?*
>
> **Reply 3**: Indeed, in the CV research community, a well-known work *[NIPS 2020] [BYOL] Bootstrap Your Own Latent* uses two neural networks, i.e., online and target networks. It trains the online network to predict the target network’s representation of another augmented view of the same image,  without using negative examples. Differently, in this work, our goal is to avoid using negative examples as well as positive examples, i.e., view augmentation. By performing structural clustering in the graph space, LPA can effectively exploit the structural information naturally present in graph data, which is free. The resulting cluster labels serve as strong self-supervision signals to guide the pre-training process.
>
> ---
>
> **Point 4**: *What is the link weight and how to obtain these weights?*
>
> **Reply 4**: Link weights are also called edge weights. They are typically described by an adjacency matrix. For example, for the adjacency matrix $A$ between "Author" objects and "Paper" objects, $A_{ij}=1$ if author $i$ writes paper $j$, i.e., the weight is 1 for the link $<i,j>$.
>
> ---
>
> **Point 5**: *How to obtain the attention coefficients? Are they model parameters?*
>
> **Reply 5**: In general, attention coefficients are computed based on both the input data and the model parameters. Therefore, attention coefficients are parametrized by model parameters. In specific, different Att-HGNN encoders have different ways to compute attention coefficients. For the adopted ie-HGCN encoder, it computes attention coefficients by inputting the projected features into a small parametrized neural network.
>
> ---
>
> **Point 6**: *In object classification, is it fair to use the object embeddings learned by semi-supervised baselines for comparison? The paper should compare the results of unsupervised methods with the results obtained by the classifiers of semi-supervised methods.*
>
> **Reply 6**: We are sorry for leading you to this confusion. Yes, for semi-supervised methods, we directly report the classification results obtained by their own classifiers. According to your comment, we have updated our manuscript to make it clearer. Please see Section 5.4 (Lines 264-265).

---

> > ### Comment · Reviewer_1dEZ · 2022-08-08
> > **Rely to the author rebuttle**
> >
> > Thanks to all the authors for their detailed replies to my questions. Your reply have mostly answered my questions. I will keep my positive score.

---

### Official Review · Reviewer_cQLi · 2022-07-13

**Rating:** 5
**Confidence:** 5
**Soundness:** 2 fair
**Presentation:** 2 fair
**Contribution:** 2 fair

**Summary:**

In this paper, the authors propose to leverage structural clustering for pre-training heterogeneous graph neural networks in a self-supervised manner. The proposed SHGP framework adopts Attentive HGNN as the base encoder and utilizes label propagation for structural clustering, thereby deriving pseudo labels for  self-supervised pre-training. The experiments are conducted on several benchmark datasets of heterogeneous graphs. The experimental results show that the proposed SHGP outperforms several conventional baseline methods in both tasks of object classification and clustering.

**Questions:**

* The concept of considering clustering for graph neural networks is very similar to the concept of pooling. For example, [c,d] also learn a soft cluster assignment for enhancing model learning. The authors should compare with them and emphasize the motivation of the paper.

    * [c] Ying, Z., You, J., Morris, C., Ren, X., Hamilton, W., & Leskovec, J. (2018). Hierarchical graph representation learning with differentiable pooling. Advances in neural information processing systems, 31.
    * [d] Ranjan, E., Sanyal, S., & Talukdar, P. (2020, April). Asap: Adaptive structure aware pooling for learning hierarchical graph representations. In Proceedings of the AAAI Conference on Artificial Intelligence (Vol. 34, No. 04, pp. 5470-5477).

* The whole framework combines two established components from other studies, but the key idea is still about structural clustering that is unnecessarily relevant to heterogeneous graphs. I wonder if the idea can be adopted with models for homogeneous graphs.

*Some state-of-the-art pertaining methods for both general and heterogeneous graphs [a, b], as well as some unsupervised representation learning methods [c], are not discussed and compared in the experiments. It would be better to compare with the state-of-the art methods in the experiments.

**Limitations:**

The authors do not mention any limitations and potential negative societal impact of the work. However, graph neural net related algorithms could still have some risks about resulting some abuse scenarios on social networks.

**Strengths And Weaknesses:**

Strengths
* Simple but effective framework.
* Publicly available benchmark datasets and released implementations for the reproducibility.


Weaknesses
* The idea of using clustering and pooling to benefit graph neural nets is not novel.
* Do not propose novel methods directly related to heterogeneous graphs.
* Some state-of-the-art pertaining methods for both general and heterogeneous graphs [a, b], as well as some unsupervised representation learning methods [c], are not discussed and compared in the experiments.

    * [a] Chien, E., Chang, W. C., Hsieh, C. J., Yu, H. F., Zhang, J., Milenkovic, O., & Dhillon, I. S. (2021, September). Node Feature Extraction by Self-Supervised Multi-scale Neighborhood Prediction. In International Conference on Learning Representations.
    * [b] Wang, X., Liu, N., Han, H., & Shi, C. (2021, August). Self-supervised heterogeneous graph neural network with co-contrastive learning. In Proceedings of the 27th ACM SIGKDD Conference on Knowledge Discovery & Data Mining (pp. 1726-1736).
    * [c] Jiang, J. Y., Li, Z., Ju, C. J. T., & Wang, W. (2020, October). Maru: Meta-context aware random walks for heterogeneous network representation learning. In Proceedings of the 29th ACM International Conference on Information & Knowledge Management (pp. 575-584).

* No theoretical supports.

---

> ### Author Response · Authors · 2022-08-02
> **Response to Reviewer cQLi**
>
> Thanks for your comments. In the following, we respond to your concerns point by point.
>
> ---
>
> **Point 1**: *The concept of using clustering for GNNs is similar to the concept of pooling, e.g. [d,e].*
>
> **Reply 1**: Graph pooling methods such as [d,e] learn a soft cluster assignment in each graph pooling layer. Then, nodes are mapped into a set of clusters, forming a coarsened graph that serves as the input of the next GNN layer. Differently, our method propagates integer (hard) cluster labels in each layer but does not coarsen the graph topology. Our goal is to obtain cluster labels through performing structural clustering. According to your suggestion, we have discussed and cited them [d,e] in our updated manuscript. Please see the last paragraph of Section 2 (Lines 95-97).
>
> ---
>
> **Point 2**: *If the idea can be adopted with models for homogeneous graphs?*
>
> **Reply 2**: In this work, like most existing related works, we aim to propose a general self-supervised pre-training framework. As a general framework, it can be applied to homogeneous graphs by establishing (homogeneous) GNN encoders, but in this paper, we focus on heterogeneous graphs which are more challenging, as we explained this in Section 1 (Line 55). In future work, we would like to apply our idea to homogeneous graphs.
>
> ---
>
> **Point 3**: *Some related works [a,b,c] are missing.*
>
> **Reply 3**: First, we would like to kindly remind the reviewer that this comment involves one **factual error**. For reference [b], it was cited as [29] in our original manuscript (cited as [33] in the updated manuscript). Specifically, we discussed it in Section 1 (Line 41) and Section 2 (Line 76). Moreover, we also considered it as one of the baselines in our experiments, i.e., HeCo. Please see Section 5.2 (Line 236). For reference [a], it focuses on the problem of extracting numerical node features from raw data, under graph-structured self-supervision. Reference [c] learns object embeddings by exploiting meta-contexts in random walks, which belongs to traditional network embedding approaches. According to your suggestion, we have added the discussion and citation of them [a,c] in our updated manuscript. Please see the last paragraph of Section 2 (Lines 92-94).
>
> ---
>
> **References**:
>
> [a] [ICLR 2022] Node Feature Extraction by Self-Supervised Multi-scale Neighborhood Prediction
>
> [b] [KDD 2021] Self-supervised Heterogeneous Graph Neural Network with Co-contrastive Learning
>
> [c] [CIKM 2020] MARU: Meta-context Aware Random Walks for Heterogeneous Network Representation Learning
>
> [d] [NeurIPS 2018] Hierarchical Graph Representation Learning with Differentiable Pooling
>
> [e] [AAAI 2020] ASAP: Adaptive Structure Aware Pooling for Learning Hierarchical Graph Representations

---

> ### Author Response · Authors · 2022-08-09
> **We Have Addressed Your Minor Concerns**
>
> Dear Reviewer cQLi,
>
> We sincerely appreciate your comments. In these comments, you pointed out three minor concerns: (1) the difference between the concept of graph pooling; (2) whether the idea can be applied to homogeneous graphs; (3) the lack of discussion of several related works.
>
> In the rebuttal text, we have made detailed explanations to address your concerns, and we have also updated our manuscript accordingly.
>
> Considering that we have addressed these minor concerns, could you please be positive about of work?
>
> We believe that our work makes a meaningful contribution to the research community, since it is the first work to emancipate the HIN-oriented SSL research from the tedious tuning of strategies for constructing positive examples and negative examples.
>
> Thank you very much!
>
> Best Regard,
>
> The Authors

---

> > ### Comment · Reviewer_cQLi · 2022-08-10
> > **Thanks**
> >
> > Thanks for addressing my question. Based on the authors' response, I increased my score from 4 to 5 accordingly.

---

> > > ### Author Response · Authors · 2022-08-10
> > > **Thank You Very Much**
> > >
> > > We really appreciate your acknowledgment of our work and your valuable comments which help improve the quality of our manuscript.

---

### Author Response · Authors · 2022-08-08
**Looking forward to the feedback of reviewers**

Dear Reviewers,

We sincerely appreciate your constructive and helpful comments. During the Author Rebuttal period, we have carefully read your reviews and made our best efforts to address all the concerns described in both the weaknesses and the questions.

As the Author-Reviewer Discussion period ends soon, we are looking forward to your feedback. Could you please let us know if our work requires any further improvement? It would be highly appreciated if you could raise the rating score of our work, considering that we have addressed your current concerns.

Thanks for your valuable time and efforts in reviewing our manuscript.

Best Regards,

The Authors

---

### Meta-Review · Area_Chair_hJWC · 2022-08-29

**Recommendation:** Accept
**Confidence:** Certain

**Metareview:**

I recommend to accept this paper.

In this paper, the authors propose SHGP, a technique to leverage structural clustering for pre-training heterogeneous graph neural networks in a self-supervised manner. After the rebuttal, all the reviewers are positive to accept this paper. I would encourage the authors to revise the paper based on the suggestions from reviewers.

**Award:**

No

---

### Decision · Program_Chairs · 2022-09-14

Accept